# Microstructure Evolution and Mechanical Properties of 20%SiC_p_/Al Joint Prepared via Laser Welding

**DOI:** 10.3390/ma15176046

**Published:** 2022-09-01

**Authors:** Fei Li, Yiming Jiang, Gaoyang Mi, Mingyang Zhang, Chunming Wang

**Affiliations:** 1Chengdu Aircraft Industrial (Group) Co., Ltd., Chengdu 610092, China; 2School of Materials Science and Engineering, Huazhong University of Science and Technology, Wuhan 430074, China

**Keywords:** laser welding, 20 vol% SiC/2A14 Al alloy, microstructure, tensile property

## Abstract

SiC particles-reinforced Al matrix composites (SiC_p_/AMCs) have been widely used in the aerospace structural components. In this work, 20 vol% SiC_p_/2A14 joint was fabricated by laser welding technology. The effects of different laser power/welding velocity on the 20 vol% SiC_p_/2A14 joint forming, microstructure evolution and mechanical properties were studied in detail. The results showed that, under the same heat input, the high power/high welding velocity was beneficial to reduce the porosity of SiC_p_/2A14 joint and inhibited the formation of brittle phase of Al_4_C_3_. At 8 kW-133 mm/s welding parameters, the maximum tensile strength of the SiC_p_/2A14 joint reached 199 MPa, which is ~64% higher than that of the SiC_p_/2A14 joint prepared at 4 kW-66 mm/s welding parameters. By analyzing the fracture morphology and SEM image of SiCp/2A14 joint section, it is was found that the porosity of weld and Al_4_C_3_ brittle phase were the important factors limiting the strength of SiCp/2A14 joint. This work provides a reference for the process window design of laser welding SiC_p_/2A14 composites.

## 1. Introduction

Today, energy conservation and emission reduction is the goal of unremitting efforts in the world, which is one of the requirements in the field of materials is to achieve the preparation and processing of light and high strength alloys [1,2]. As an advanced material integrating structure and function, SiC particles-reinforced Al (SiC_p_/Al) can be widely used in the field of aerospace structural components [3,4]. The joining of alloys is an indispensable technology in the current manufacturing process, and the study of the welding behavior of new SiC_p_/Al alloy is the key to further application in the field of aerospace [5,6]. Examilioti et al. systematically designed a series of welding process parameters to fully understand the welding characteristics of AA2198. Its design process parameters are laser power increased from 5 kW to 8 kW. They reported that the combination of high power and high welding speed is conducive to the formation of type I welds, thus improving the joint strength [7].

It is well known that the phase interface of metal matrix composites plays an important role in structural and functional properties [8,9,10]. Unfortunately, SiC_p_/Al is recognized as one of the most challenging fusion welding materials due to deterioration of SiC_p_/Al joint properties caused by interfacial reaction between SiC particles and Al matrix (4Al + 3SiC → 3Si + Al_4_C_3_) [11]. In the field of fusion welding, the most effective strategy was to add alloy foil to suppress the volume fraction of brittle phase of Al_4_C_3_, so as to improve the strength of SiC_p_/Al joint [12,13,14]. However, the strategy of adding alloy foil had two major shortcomings. One is that the alloy foil usually reacted with aluminum to form large brittle phases such as Al_3_Ti and Al_3_Zr, which reduced the toughness of the SiC_p_/Al joint [12,14]. Second, the foil layer was difficult to evenly distribute in the fusion zone. The unmelted alloy foil layer was concentrated in the center of the fusion zone [12]. With the development of laser technology, the laser power, spot size and spot energy distribution have been greatly studied and developed, which provided a new possibility for laser welding SiC_p_/Al composites [15,16]. Hence, it will provide new understanding to seek the process strategy of laser welding SiC_p_/Al composites, which is related to whether the reaction degree of SiC can be reduced by adjusting the welding process parameters.

This work cleverly controls the reaction degree of SiC by controlling the residence time of molten pool. Under the same heat input, the process parameters of different laser welding power/welding velocity were designed. This work carefully characterized the microstructure of the 20 vol% SiC_p_/Al joint, and the morphology evolution mechanism of SiC during high laser power/welding velocity was revealed in detail. This work provides a new insight into the inherent relationship between the microstructure and mechanical properties of the 20 vol%SiC_p_/Al joint.

## 2. Materials and Methods

T6 condition-SiC_p_/2A14Al composites with volume fraction of 20 vol% SiC particles was selected as the base material (Figure 1). The 20 vol% SiCp/Al was purchased from Shenzhen Superior Technology New Material Co., LTD. Detailed information about SiC_p_/Al composite materials are as follows: Cu 3.9–4.8, Mn 0.4–1.0, Si 0.6–1.2, Mg 0.4–0.8, Zn ≤ 0.3, Ti ≤ 0.15, Ni ≤ 0.1, Al bal (wt. %). The size of SiC_p_/2A14 Al base material was 100 mm × 30 mm × 4 mm.

The laser beam welding system was composed by an IPG YLS-30000 fiber laser system with a Kuka IRB-6400 robot. The size of the laser spot is 500 μm. The 100% Ar was used as shielding gas during the welding process. The processing parameters for laser butt welding are shown in Table 1.

A sample with a size of 10 mm × 5 mm × 4 mm was cut via wire electric discharge machine. The metallographic samples were prepared by automatic grinding and polishing machine (EcoMet 250, BUEHLER, USA) according to the standard sample preparation process. SEM was performed using a Gemini SEM300 scanning electron microscope (SEM) manufactured by Carl Zeiss (Oberkochen, Germany). 

Microhardness of SiC_p_/2A14Al joints were measured by HVS-1000A Vickers hardness machine (Laizhou, China) with a test load of 300 g and a dwell period of 15 s. Tensile specimens with the gauge length of 15 mm, gauge width of 6 mm and thickness of 4 mm were machined parallel to the transverse direction. Uniaxial tensile tests for each specimen were performed using a high temperature endurance testing machine (Autograp AG-IC 100 kN, Shimadzu, Japan) with a constant strain rate of 2 mm/min at ambient temperature. Each group of tensile experiment was repeated two times.

## 3. Results

### 3.1. Microstructure Characterization

Figure 2 shows the SEM images of the welded joint cross section. It can be seen that even under the same nominal heat input (60 J/mm), the welded joint cross-section porosity shows distinct differences. Sample Y1 showed a large number of pores, and the pores were densely distributed in the upper and lower areas of the weld. When the laser welding power was 8000 W-133 mm/s, the number of pores in sample Y2 decreased obviously. In the metallurgical reaction process of welding molten pool, the temperature and cooling rate of the molten pool control the growth morphology of microstructure and the reaction degree of SiC particles (the preferred growth factor of intrinsic crystal structure of the phase is not considered here temporarily). Due to the presence of SiC in aluminum matrix, the viscosity of molten pool increased with the increase of SiC content in the welding process, which increases the difficulty of gas escape in the welding process [17,18]. When the welding power and welding velocity were changed at the same heat input, the porosity of the joint cross section showed great difference, as shown in Figure 2. The melt viscosity of SiC_p_/2A14 metal matrix composites was also affected by the volume fraction of SiC particles. A large number of solid particles in the molten pool obstruct the flow in the welding process. During laser welding, the plasma pressure makes the fluid in the molten pool move violently, which has an important influence on the redistribution of the enhanced phase in the molten pool. Therefore, the increase of laser power may increase the degree of molten pool fluid reaction and form a penetrating deep fusion welding mode. At the same time, stomatal escape channels are more favorable, resulting in lower stomatal rate.

Figure 3 shows the typical microstructure images of the SiC/2A14 joint, which could be roughly divided into α-Al matrix, Al-Cu phase and long rod Al_4_C_3_ brittle phase. Furthermore, it can be seen from the distribution diagram of element in Figure 3c that some Si elements were distributed in disorder and the other in eutectic phase. The carbon atoms provided by the dissolution of SiC reacted with α-Al first, and Si atoms were distributed in the form of solid solution and compound formation such as Mg_2_Si.

For laser welding of particle-reinforced aluminum matrix composites, the degree of reaction between SiC particles and aluminum matrix in welding molten pool and the reaction products are important factors to construct the microstructure of fusion zone. In order to better reflect the reaction behavior of SiC particles in the molten pool, Figure 4 shows the SEM morphology of SiC particles in the fusion zone. As can be seen from Figure 4, the surface of SiC particles was decomposed to varying degrees. Long rods of Al_4_C_3_ are distributed around SiC. However, it is worth noting that the formation path of acicular Al_4_C_3_ is complex, which is mainly related to the thermal environment of the welding molten pool. For the design strategy of this work, the combination of two sets of process parameters with different laser power/welding velocity created a molten pool with two different thermal conditions, which determined the reaction time of SiC particles in the molten pool. In the laser welding process, the area formed by the keyhole had a relatively high temperature field. The SiC particles subjected to the keyhole also have the strongest dissolution and even decomposition effect. In the molten pool formed near the keyhole, SiC was retained in the molten pool due to its high melting point (2730 °C) without direct laser beam under ideal assumption, but the dissolution reaction of SiC will continue. That is to say, the energy of the keyhole (laser power/laser spot size) determines the melting behavior of part of the SiC, and the retention time of the molten pool per unit length determines the SiC dissolution behavior.

Figure 5 shows the characteristic morphology and dissolution behavior of SiC in the fusion zone of Y2 joint. From the SEM morphology, it can be clearly seen that the dissolution behavior of SiC particles occurs at the edge of SiC particles. The dissolution behavior was synchronized with the precipitation behavior. It can be explained that the SiC particles begin to dissolve at the edge of the particle due to the influence of high temperature in the molten pool, and the dissolved products Si and C react rapidly with liquid Al. In addition, a large number of carbon atoms around SiC particles were rapidly consumed. However, Si atoms cannot be consumed well, leading to local Si enrichment, which slows down the dissolution and precipitation behavior to a certain extent [19,20]. Si element is the product of the reaction; the addition of Si element will reduce the reaction of the interface reaction rate, effectively inhibiting Al_4_C_3_ production, but also significantly improve the interface wettability. Ti doping can significantly improve the separation of interfaces with different configurations. However, Mg and Cu doping can terminate the interface binding ability of Al/Si, but C and SiC interface separation effectivity is reduced, so the enhancement effect of interface bonding is limited [21,22,23,24]. 

For the dissolution and precipitation behavior of particles, the interface products at the particle interface was an interesting discussion. For the SiC particles in the molten pool, the unmelted SiC tends to form Al-Si-C ternary reaction products in the form of dissolution and precipitation. Although Al_4_C_3_ is prone to hydrolysis reactions that reduce the mechanical properties of the SiC_p_/Al joint, some studies have shown that the nano-scale Al_4_C_3_ was conducive to load transfer and enhanced the mechanical properties of the alloy. In the laser welding process, SiC_p_/Al particles dissolved in the molten pool with the melting of aluminum matrix. Therefore, the microstructure of the weld was essentially the aluminum matrix reinforced by ex situ SiC particles. Compared with the in situ formation particles-reinforced aluminum matrix, the cracks usually initiate and expand rapidly in the stress concentration region on the particle surface [25,26]. Therefore, the bonding strength of the interface between the particles and matrix determines the mechanical properties of the composites [27]. The interfacial bonding strength between SiC particles and aluminum matrix can be improved by regulating in situ Al-Si-C ternary products around SiC particles.

### 3.2. Microhardness and Tensile Strength

Figure 6 shows the microhardness curves of the cross-section of the joint. The distance between adjacent microhardness points was 0.2 mm. As can be seen from Figure 6, due to the secondary rearrangement of SiC, porosity and Al_4_C_3_, the joint microhardness presents a trend of uneven change.

Figure 7 shows the tensile strength curve of the welded joint. The combination of ‘laser power-welding velocity’ strongly affected the tensile strength of the welded joints. For the laser welding process parameters of 4 kW-66 mm/s, the tensile strength of the welded joint Y1 was 121 MPa, while the tensile strength of the welded joint Y2 prepared by the welding process parameters of 8 kW-133 mm/s was 199 MPa. In general, the nominal tensile strength of the joint depends on two factors: weld formation and microstructure. Among them, welding forming includes whether the joint has defects such as edge bite and welding porosity, as shown in Figure 2. The microstructure of the joint is the key factor to determine the mechanical properties of the joint. For the 20%SiC/2A14 joint, the microstructure region could be roughly divided into fusion zone, heat affected zone and base zone. A needle-like brittle Al_4_C_3_ phase appeared in the fusion zone of the weld. The SiC enrichment appeared between the heat-affected zone and the fusion zone, both of which did not occur in the base material, as shown in Figure 2.

In order to better reflect the internal relationship between microstructure and mechanical properties, the tensile fracture morphology of the joint is shown in Figure 8. Without exception, the SiC reaction products observed in the cross section of the weld are needle-like/long rod shape, while a large number of step-like phases are observed in the fracture. It can be reasonably inferred that the morphology and distribution orientation of Al_4_C_3_ were not formed randomly. Al_4_C_3_ were determined by external thermal environment factors such as molten pool temperature gradient and intrinsic crystal growth structure factors of Al_4_C_3_.

Surprisingly, the fracture morphology of both joints showed a large number of steps, and the combination of low power/low welding velocity (4 kW-66 mm/s) process parameters showed more fracture steps. Therefore, according to the SEM morphology shown in Figure 2, the real morphology of two-dimensional acicular Al_4_C_3_ should be flake and fracture parallel to the flake surface during fracture, which also explains the origin of numerous stepped morphology in the tensile fracture.

## 4. Discussion

Based on the above results, the welded joint tensile strength was greatly improved under the process strategy of controlling the combination of laser welding parameters, and its internal control factors need to be further understood. These are as follows:

(i) Influence of welding process parameter on tensile strength. For the laser welding SiC particle-reinforced aluminum matrix composites, the welding difficulty not only has serious interface reaction difficulty between SiC particles and Al matrix, but also, the introduction of SiC particles aggravate the formation of porosity in laser welding aluminum alloy. In this study, a large number of pores appeared in the welding of 20 vol% SiC/2A14 composites, mainly distributed at the upper and lower ends of the weld cross-section. Although the number of pores was effectively reduced by optimizing the process parameters, sample Y2 still showed unsatisfactory porosity, as shown in Figure 2.

(ii) Influence of morphology of Al_4_C_3_ on tensile strength. In general, the growth of Al_4_C_3_ was related to the inter-diffusion of Al and C atoms, and the growth rate (*dw*/*dt*) of Al_4_C_3_ could be expressed based on Arrhenius’s law Equation (1),
(1)$$\frac{dw}{dt}=\sqrt{\frac{k}{2t}\exp\left(-\frac{Q}{RT}\right)}$$
where *k*, *w*, *Q*, *R*, *t*, and *T* are the constant, weight of Al_4_C_3_, activation energy, gas constant, holding time, and temperature, respectively [28,29,30,31]. As a result, the growth rate of Al_4_C_3_ increases with temperature increasing, and decreases with holding time prolonging [28].

Jiang et al. investigated the nucleation and growth mechanism of aluminum carbide in graphene nanosheet/Al composites. They found that the longitudinal growth interface Al_4_C_3_ grew faster along the (003) plane than along the (003) plane, most of which grew into rods [28]. Since the longitudinal growth of Al_4_C_3_ was controlled by diffusion, there are C and Al in both Al_4_C_3_ and Al matrix, while the lateral growth of Al_4_C_3_ was affected by the nucleation rate at the interface [32]. The rhomboid structure (R-3M) of Al_4_C_3_ consists of superposed layers of Al_2_C and Al_2_C_2_ [33]. In the Al_2_C layer, C atoms were tightly packed in the octahedral void of Al atoms, but not tightly packed in the Al_2_C_2_ layer. Thus, transverse diffusion of C atoms through the Al_2_C_2_ layer was expected, since the Al_2_C layer would block normal diffusion [34]. Thus, the growth of Al_4_C_3_ along the X direction was determined by the diffusion of C atoms in the Al_2_C_2_ layer, while the growth along the Y direction was determined by the alternate nucleation of Al_2_C_2_ and Al_2_C layers [28]. The difference in growth rates between the two directions eventually resulted in the rod-like appearance of Al_4_C_3_. Based on the above analysis, Al_4_C_3_ phase tended to nucleate at the opening edge of SiC due to its high chemical reactivity. Thereafter, the growth of Al_4_C_3_ was controlled by its crystal structure characteristics, which determine the growth and morphology of Al_4_C_3_ [35]. 

Figure 9 shows the schematic diagram of two Al_4_C_3_ in laser welded SiCp/2A14 composites. Similarly, it can be seen from Figure 5 and Figure 8 that the morphology of Al_4_C_3_ can be subdivided into two types. One is complete Al_4_C_3_, which has a sheet shape. The other is the non-complete Al_4_C_3_ phase around SiC particles. Therefore, it is reasonable to speculate that the growth morphology of Al_4_C_3_ was closely related to the temperature field of molten pool. The cooling rate of the molten pool near the keyhole was relatively slow, which was conducive to the full nucleation and growth of Al_4_C_3_. In the area near the fusion line, due to the faster cooling rate and lower temperature field, the nucleation and growth behavior of Al_4_C_3_ cannot be fully carried out. So, the growth of incomplete Al_4_C_3_ was found around the SiC.

## 5. Conclusions

In this study, the 20 vol% SiC/2A14 composites were systematically studied for laser welding. Based on the SEM characterizations and fracture behavior analysis, the main conclusions can be summarized as follows:

(1) Under the same heat input 60 J/mm, the process parameters of high power/high welding rate were beneficial to reduce joint porosity. The increase of laser power was beneficial to improve the viscosity of molten pool. The increase of welding rate was beneficial to reduce the dissolution reaction time of SiC, which synergically improved the joint strength.

(2) Compared with the welding parameters of 4000 W-67 mm/s, the tensile strength of the joint prepared with 8000 W-133 mm/s welding parameters can reach ~199 MPa, which is ~64% higher. The brittle phase of Al_4_C_3_ formed during laser welding was the key factor restricting the strength of the joint, and its distribution pattern further reduces the strength of the joint.

(3) During laser welding, the plasma pressure makes the fluid in the molten pool move violently, which has an important influence on the redistribution of the enhanced phase in the molten pool. The increase of laser welding power form a penetrating deep fusion welding mode. At the same time, stomatal escape channels are more favorable, resulting in lower stomatal rate.

(4) The morphology of Al_4_C_3_ can be subdivided into two types. One is complete Al_4_C_3_, which has a sheet shape. The other is the non-complete Al_4_C_3_ precipitation around SiC particles. The cooling rate of the molten pool near the keyhole is relatively slow, which is conducive to the full nucleation and growth of Al_4_C_3_.

## Figures and Tables

**Figure 1 materials-15-06046-f001:**
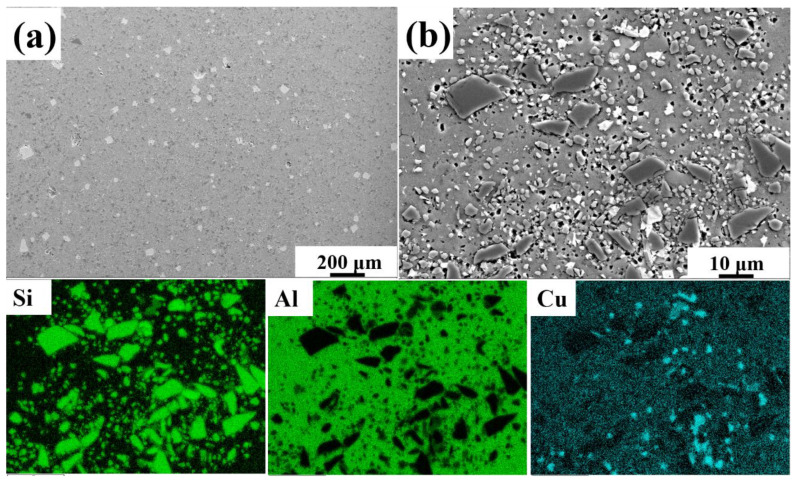
SEM images of 20 vol%SiC/2A14 base material and corresponding surface elements distribution mapping. (**a**) SEM image of base material with low magnification; (**b**) SEM image of characteristic region of base material.

**Figure 2 materials-15-06046-f002:**
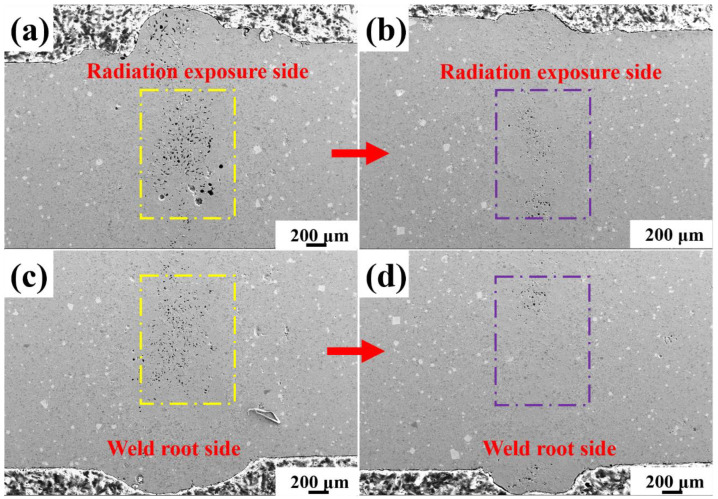
SEM image of 20 vol%SiC/2A14 joint cross section. (**a**,**c**) Representative sample Y1; (**b**,**d**) Representative sample Y2.

**Figure 3 materials-15-06046-f003:**
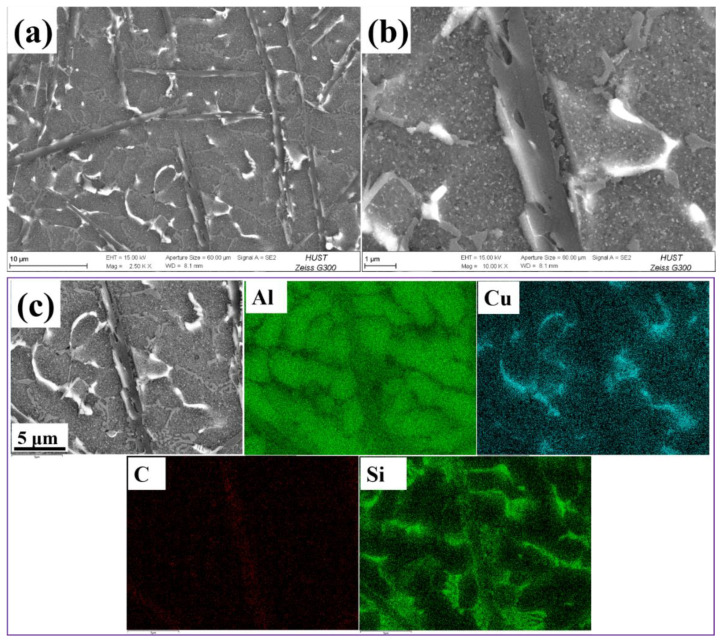
SEM image and elements mapping of 20 vol%SiC/2A14 joint cross section. (**a**–**c**) Representative sample Y1.

**Figure 4 materials-15-06046-f004:**
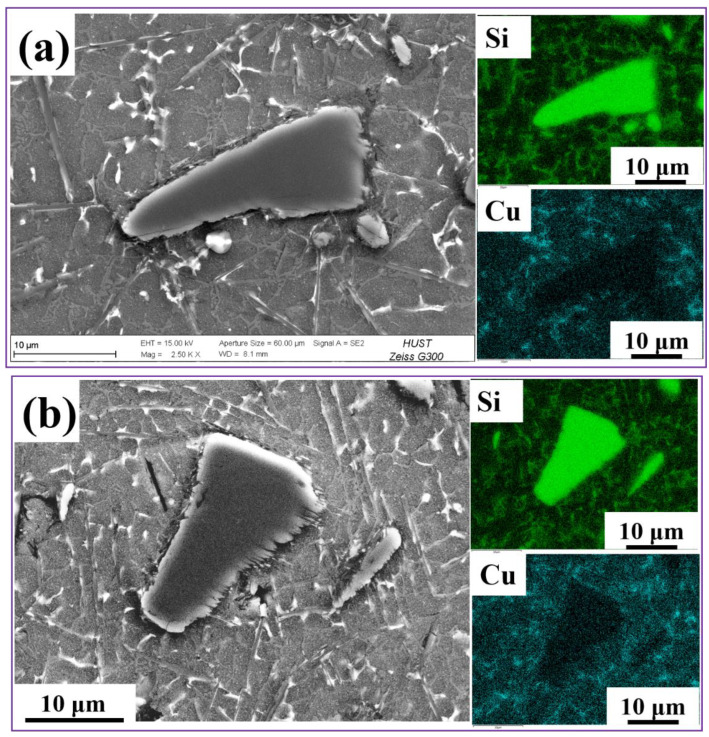
SEM image of SiC in the fusion zone. (**a**) Representative sample Y1; (**b**) Representative sample Y2.

**Figure 5 materials-15-06046-f005:**
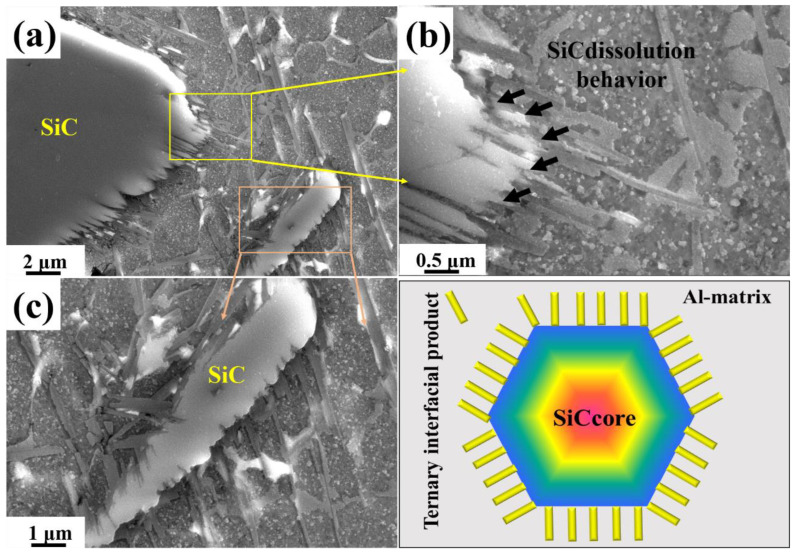
Characteristic SEM image of SiC and schematic diagram of its dissolution behavior in the sample Y2. (**a**) Microstructure near dissolved SiC in a weld; (**b**,**c**) SEM detail image of feature region in (**a**); (**c**) Schematic diagram of SiC dissolution.

**Figure 6 materials-15-06046-f006:**
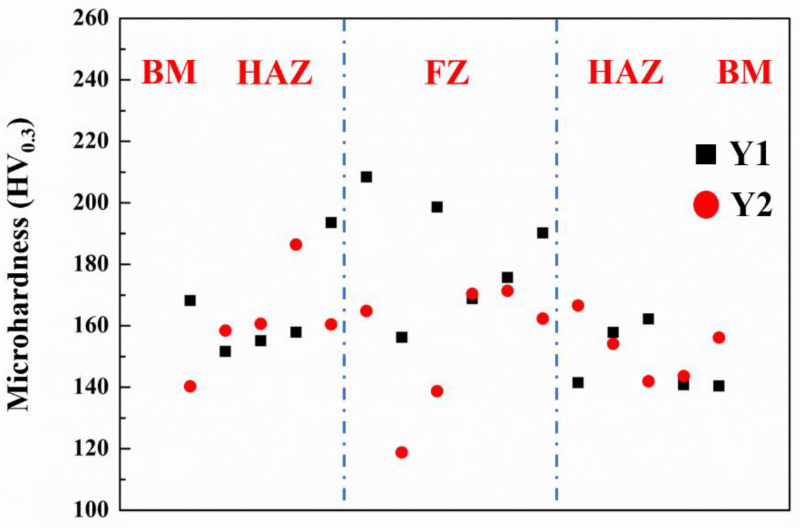
Microhardness curves of two samples.

**Figure 7 materials-15-06046-f007:**
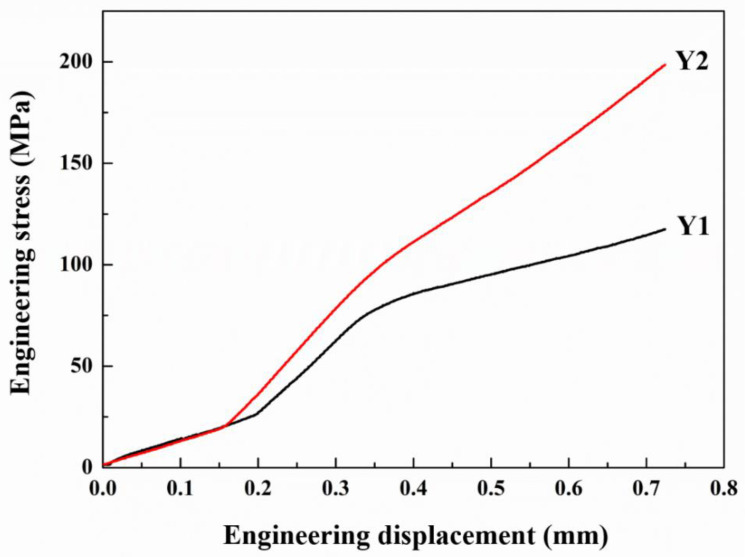
Engineering stress-engineering displacement curves of two samples.

**Figure 8 materials-15-06046-f008:**
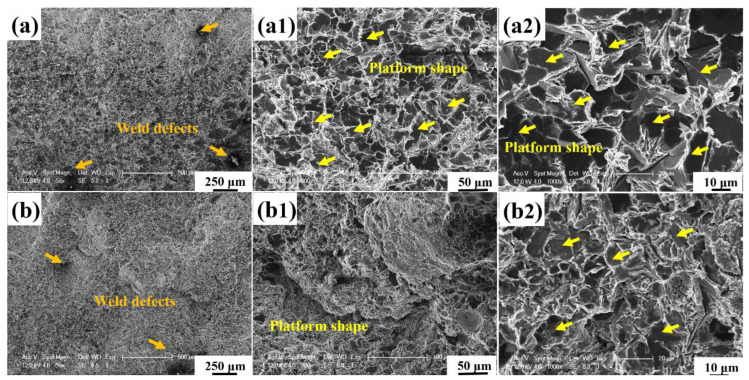
SEM characteristic images of fracture morphology of two groups of joints. (**a**–**a2**) Representative sample Y1; (**b**–**b2**) Representative sample Y2.

**Figure 9 materials-15-06046-f009:**
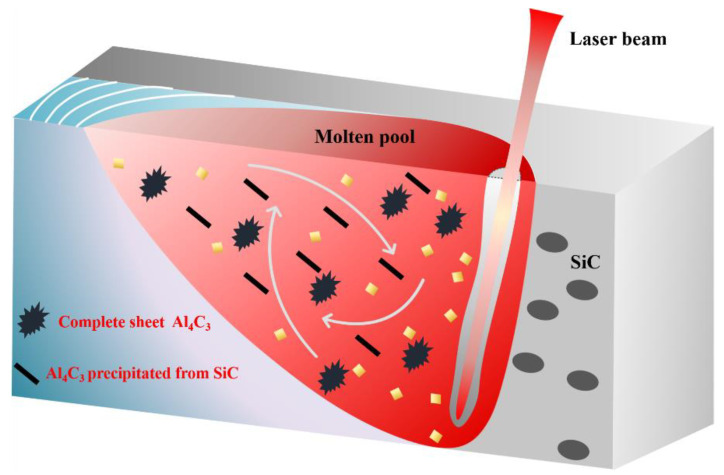
The schematic diagram of two Al_4_C_3_ in laser welded 20 vol% SiCp/2A14 composites.

**Table 1 materials-15-06046-t001:** Laser welding parameters.

Samples	Laser Powder	Welding Velocity
Y1	4000 W	67 (mm/s)
Y2	8000 W	133 (mm/s)

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
