# Peer review of "Microstructure Evolution and Mechanical Properties of 20%SiCp/Al Joint Prepared via Laser Welding"

_materials, 2022, doi:10.3390/ma15176046_

Round 1

Reviewer 1 Report

1.     Is there any quantitative analysis about intermetallic compounds to compare between 4000W and 8000W, please provide them with a correlation with mechanical properties.

2.     In conclusion, please add one more point about the porosity formation and reduction mechanism for both types of welding conditions.

3.     Is the vaporization of materials causes to form any porosity due to laser power justify?

4.     According to fig 8, what is the molten pool length or solidification rate to identify the formation of the compound.

5.     Figure 7 title should need to add sub-figure names.

6.     What is the largest Si particles were formed and their distribution through the welds needs to be quantified as a area-wise.

7.     What is the hardness of the SiC particles compared to the matrix, please provide hardness measurement plots for both the welds.

Reviewer 2 Report

1. It is not clear why the authors made a sharp transition from 4000 watts to 8000 watts. With a constant, as I understand it, spot diameter. Let them explain.

2. Does the composition of the melt affect the dissolution of silicon carbide (titanium, nickel, zinc, that alloy content). And how do they influence? In the case of the presence of these elements, even in small quantities, the strength of the weld can significantly deteriorate. Need an explanation!

3. There is no red dots explanation in figure 5.

4. The conclusions are very short, it is necessary to expand them 2 times.

I wish the authors success.

Reviewer 3 Report

Comments/Suggestions:

There is not significant changes have made in manuscript titled “Microstructure evolution and mechanical properties of 20%SiCp/Al joint prepared via laser welding” submitted by Fei Li et al..   I could not change my earlier decision.  I can not be recommend for publishing in the “Materials” journal

Round 2

Reviewer 2 Report

You need to add confidence intervals in Figure 6.

Since there are no data on studying the dimensions of the melt bath with such a sharp change in the laser power, it is necessary to cite literature data on the relationship between the geometric parameters of the bath and the energy parameters of the laser.

The authors state that titanium, zinc and nickel impurities are present in the alloy used, but gently "deny" their influence on the solubility of silicon carbide and their influence on the properties of the weld - this is unacceptable and requires clarification. Explain this effect (of influence on solubility) by referring to data from the literature. (3-4 sentences).

I am sure that the authors will respond to the above comments.

Best regards!

Author Response

Responses to Reviewers’ comments

Reply to reviewer 2

You need to add confidence intervals in Figure 6.

Response: Thank you very much for your comprehensive comment. This is a very detailed comment. For homogeneous alloys, the microhardness should be repeated three times as much as possible to ensure the accuracy of the test results. However, for particle reinforced aluminum matrix composites (especially after the welding process), each area is unique, which makes repeated testing of microhardness extremely difficult. Because of the presence of SiC particles, the indentation movement will lead to contact with the SiC particles, making the microhardness result soaring, which is unreasonable for adding error bars.

Since there are no data on studying the dimensions of the melt bath with such a sharp change in the laser power, it is necessary to cite literature data on the relationship between the geometric parameters of the bath and the energy parameters of the laser.

Response: We thank the reviewer very much for pointing out this problem. Based on your detailed comment, we have cited literature data on the relationship between the geometric parameters of the bath and the energy parameters of the laser.

 Examilioti et al. systematically designed a series of welding process parameters to fully understand the welding characteristics of AA2198. Its design process parameters are laser power increased from 5kW to 8kW. They reported that the combination of high power and high welding speed is conducive to the formation of type I welds, thus improving the joint strength [7].

[7] Theano N. Examilioti, Nikolai Kashaev, Josephin Enz, Benjamin Klusemann, Nikolaos D. Alexopoulos, On the influence of laser beam welding parameters for autogenous AA2198 welded joints, Int. J. Adv. Manuf. Technol, 110 (2020) 2079-2092.

The authors state that titanium, zinc and nickel impurities are present in the alloy used, but gently "deny" their influence on the solubility of silicon carbide and their influence on the properties of the weld - this is unacceptable and requires clarification. Explain this effect (of influence on solubility) by referring to data from the literature. (3-4 sentences).

Response: We thank the reviewer very much for pointing out this problem. Based on your detailed comment, we have cited literature data on the relationship between the geometric parameters of the bath and the energy parameters of the laser:

Si element is the product of the reaction, the addition of Si element will reduce the reaction of the interface reaction rate, effectively inhibit Al4C3 production, but also significantly improve the interface wettability. Ti doping can significantly improve the separation of interfaces with different configurations work. However, Mg and Cu doping can terminate the interface binding ability of Al/Si enhanced, but make C end SiC interface separation work reduced, so the enhancement effect of interface bonding is limited.

[21] Qiu Feng, Tong Haotian, Shen Ping, Cong Xiaoshuang, Wang Yi, Jiang Qichuan, Overview: SiC/Al interface reaction and interface structure evolution eechanism, Acta. Metall. Sin, 1 (2019) 87-100.

[22] Fang X. Thoeretical prediction of interfacial reaction and work of adhesion in SiC/Al composites. Shanghai: Shanghai Jiao Tong University, 2013

[23] Fang X, Fan T X, Zhang D. Work of adhesion in Al/SiC composites with alloying element addition [J]. Metall. Mater. Trans, 44 (2013) 5192.

[24] Lee J C, Byun J Y, Park S B, et al. Prediction of Si contents to suppress the formation of Al4C3 in the SiCp/Al composite [J]. Acta. Mater, 46(1998) 1771.

I am sure that the authors will respond to the above comments.

Best regards!

Response: Thank you very much for your comprehensive comments of the manuscript. Your comments are very instructive to our manuscript. In accordance with your professional comments, we try our best to explain and supplement the contents of the manuscript, and look forward to your approval. The following is our careful response to your comments. We hope these revisions are suitable.

Best regards!

This manuscript is a resubmission of an earlier submission. The following is a list of the peer review reports and author responses from that submission.

Round 1

Reviewer 1 Report

The manuscript titled “Microstructure evolution and mechanical properties of 20%SiCp/Al joint prepared via laser welding” submitted by Fei Li et al.is very poorly written and lacks both qualitative/ quantitative discussion and novelty (https://www.sciencedirect.com/science/article/pii/S135983682200261X). Even the interpretation of some of the observed experimental and theoretical features is written extremely poor. In general, the authors offer very shallow analysis of the experimental and theoretical results, which is effectively simplified down to labelling the ‘typical’ feature with the ‘typical’ conclusion. Because of its low quality analysis, the manuscript is unlikely to attract an interest of the journal readers. While considering all the above mentioned factors I do not recommend this manuscript for publishing in the “Materials” journal since it seems that no novelty is reported in the present manuscript. 

Reviewer 2 Report

There are comments and suggestions to the article:

1. It is not clear how the original composite was obtained.

2. There is no information about the structure and composition of the material after cutting, grinding. There is no data on whether the surface was cleaned after machining.

3. It is not clear why the authors made a sharp transition from 4000 watts to 8000 watts.

4. What was the size of the laser spot.

5. It is necessary to describe Figure 2 in more detail by making references to it in the text.

6. Does the composition of the melt affect the dissolution of silicon carbide (titanium, zirconium, which were discussed above).

7. How the main parameters of laser welding affect the amount of intermediate phase formed. It is necessary to evaluate the kinetics of silicon carbide dissolution in more detail by increasing the number of experiments, smoothly changing the deposition parameters, and estimating the size and amount of the intermediate phase.

8. Why compare a more porous starting material with a less porous one with such a sharp change in parameters?

9. There is no wavy line in figure 5.

10. It is not clear why the velocity equation is given if there are no theoretical estimates of it.

11. Is it possible to cite literature data on how growth from the surface of silicon carbide occurs, are there such studies?

I wish the authors of the article to calmly accept criticism, calmly finalize the article and release high-quality material.

Reviewer 3 Report

1. Please quantify the pores and makes the relation between pores and laser power to estimate the pores for the selected alloy according to power.

2. In figure 4, microbars are mandatory for the EDS mapping images.

3. In figure 8, please indicate the things appearing with different shapes and their role on the formation of porosity.

4. What is the hardness of the brittle phase Al4C3 formed during solidification and how it is different from the weld zone and other matrix.

5. What is the reason to choose a selected range of laser power and single heat input to evaluate the composite weld ability, justify?

6. in page 8, what is the equation number and explanation with units is needed for each and individual characters.

7. Is there any grain size variation due to changes in laser power and mechanical properties, clarification is needed.